# Cutaneous Squamous Cell Carcinoma in Immunocompromised Patients—A Comparison between Different Immunomodulating Conditions

**DOI:** 10.3390/cancers15061764

**Published:** 2023-03-14

**Authors:** Ofir Zavdy, Tara Coreanu, Dvir Yohai Bar-On, Amit Ritter, Gideon Bachar, Thomas Shpitzer, Noga Kurman, Muhammad Mansour, Dean Ad-El, Uri Rozovski, Gilad Itchaki, Shany Sherman, Limor Azulay-Gitter, Aviram Mizrachi

**Affiliations:** 1Department of Otolaryngology Head and Neck Surgery, Rabin Medical Center, Petah Tikva 4941492, Israel; 2Sackler Faculty of Medicine, Tel Aviv University, Tel Aviv 6997801, Israel; 3Institute of Oncology, Davidoff Center, Rabin Medical Center, Petach Tikva 4941492, Israel; 4Department of Plastic Surgery and Burns, Rabin Medical Center, Petah Tikva 4941492, Israel; 5Division of Hematology, Davidoff Cancer Center, Rabin Medical Center, Petach Tikva 4941492, Israel; 6Department of Dermatology, Rabin Medical Center, Petah Tikva 4941492, Israel; 7Department of Nephrology and Hypertension, Rabin Medical Center, Petah Tikva 4941492, Israel

**Keywords:** cutaneous SCC, immunosuppression, non-melanoma skin cancer, transplants, CLL, chronic kidney disease, psoriasis

## Abstract

**Simple Summary:**

Immunosuppression is strongly associated with an increased risk of developing cutaneous squamous cell carcinoma (cSCC). Compared to controls, immunosuppressed patients demonstrated Several differences were seen between immunosuppressed (IS) patients and the control group. Lower survival rates, higher recurrence rates, multiple (malignant) primary tumors, and higher rates of positive margins were seen in IS patients compared to immunocompetent controls. Among immunosuppressed patients and solid organ transplant recipients (SOTR), patients with chronic lymphocytic leukemia (CLL), chronic kidney disease (CKD), rheumatoid arthritis (RA) and psoriasis demonstrated worse outcomes compared to controls and to other immunosuppressed groups. Active surveillance and close follow-ups by both primary oncologists and dermatologists are advised in order to diagnose and treat cSCC at an early stage.

**Abstract:**

Background: Immunosuppression is strongly associated with an increased risk of developing cutaneous squamous cell carcinoma (cSCC). Studies on solid organ transplant recipients (SOTR) and chronic lymphocytic leukemia (CLL) patients have already demonstrated higher rates of aggressive cSCC tumors in these populations compared to immunocompetent controls. Studies on other immunosuppressed patient groups are scarce. This study was aimed at assessing the effects of different immunomodulating conditions on patients diagnosed with cSCC. We sought to compare the clinical features, treatments, and survival rates among the different study groups, as well as outcomes to those of immunocompetent controls with cSCC. Methods: A retrospective analysis of 465 cSCC patients, both immunosuppressed (IS) and immunocompetent controls. Etiologies for immunosuppression included SOTR, CLL, chronic kidney disease (CKD), psoriasis, rheumatoid arthritis (RA) and systemic lupus erythematous (SLE). Results: Compared to the control group, IS patients demonstrated several significant differences. These include higher rates of positive resection margins, higher recurrence rates, and multiple SCC tumors. Patients in the IS group, who were also given immunomodulating agents, demonstrated even lower survival rates. Cox regression analysis demonstrated statistically significant decreased overall survival (OS) rates for IS patients compared to the controls (OR = 1.9, *p* = 0.031). SOTR patients tend to have multiple cSCC tumors (35%), with the highest number of primary tumors compared to controls (2.54 tumors per patient on average, *p* < 0.001), but also compared to all other IS groups. The average SCC lesion size in the SOTR group was the smallest, measuring at 13.5 mm, compared to the control group and all other IS groups. Decreased survival rates were seen on Cox regression analysis compared to controls (HR = 2.4, *p* = 0.001), but also to all other IS groups. CLL patients also had the highest rates of positive margins compared to controls (36% vs. 9%, *p* < 0.01) and to all other IS groups. They were also most likely to get adjuvant or definitive oncological treatments, either radiotherapy or chemotherapy, compared to controls (36% vs. 15%, *p* = 0.02) and to other IS groups. Patients in the CKD group demonstrated the highest rates for multiple cSCC (OR = 4.7, *p* = 0.001) and the worst rates of survival on Cox regression analysis (HR = 3.2, *p* = 0.001). Both rheumatoid arthritis and psoriasis patients demonstrated the shortest disease-free survival rates (2.9y ± 1.1, 2.3y ± 0.7, respectively), compared to controls (4.1y ± 2.8) and to all other IS groups. Conclusions: Among cSCC patients, immunosuppression due to SOTR, CLL, CKD, RA, and psoriasis is associated with worse outcomes compared to controls and other IS groups. These patients should be regarded as high-risk for developing aggressive cSCC tumors. This study is the first to assess and compare cSCC outcomes among multiple IS patient groups.

## 1. Introduction

Cutaneous squamous cell carcinoma (cSCC) is the second most common skin cancer worldwide, with basal cell carcinoma (BCC) being the first [1]. There has been a constant increase in the incidence of nonmelanoma skin cancer (NMSC) over the past few decades [2]. Ultraviolet (UV) radiation is the main risk factor for the development of cSCC, with a dose-dependent accumulative effect regarding lifelong exposure [3]. Consequently, these tumors are mostly found in sun-exposed areas of the body, mostly in the head and neck region [4,5]. Other known risk factors for cSCC development include fair skin and cigarette smoking [6,7].

Immunosuppression is strongly associated with an increased risk for cSCC. This association was reported by many researchers who focused on solid organ transplant recipients (SOTR) [8,9]. Other medical causes for immunosuppression were previously reported to be associated with cSCC. Chronic lymphocytic leukemia (CLL) is the most common leukemia in the western world, with an overall lifetime risk of 0.6% [10,11]. It is associated with significant immunosuppression, which is thought to increase the risk for second primary neoplasms. Van der Straten, et al. [12] reported an increase of more than two-fold in the risk for secondary cSCC among 25,000 CLL patients, compared to the general population. Patients with CLL also exhibit an eight-fold increased risk for non-melanoma skin cancers compared to the general population [13]. Apart from CLL, other immune-modulating conditions such as psoriasis, rheumatoid arthritis (RA), and systemic lupus erythematosus were also associated with higher incidence rates of cSCC [14,15,16].

This study aimed to assess the effects of different conditions that are associated with immunosuppression on patients diagnosed with cSCC. We sought to compare the clinical features, treatments, and survival rates among the different study groups, as well as outcomes to those of immunocompetent controls with cSCC.

## 2. Materials and Methods

### 2.1. Patients

After receiving institutional review board approval (no. RMC-21-0112), we reviewed all the electronic database charts of all adult patients who were treated for cSCC of the head and neck region in our institution between 2011 and 2020. The dataset included 465 patient records, both immunosuppressed and immunocompetent. Etiologies for immunosuppression included SOTR, rheumatic, dermatological, and nephrological causes. The data collected included patients’ medical histories, treatments, and outcomes. Disease severity was assessed according to the TNM staging system by the American Joint Committee on Cancer Tumor Classification, 8th edition (AJCC 8) [17]. We searched the registry of each patient to look for evidence of perineural invasion (PNI), extracapsular extension (ECE), positive surgical margins, the existence of multiple SCC lesions, as well as locoregional and distant recurrence rates of the primary tumor. The results of the different exposure groups were compared against the immunocompetent patients in the control group.

### 2.2. Statistical Analysis

All statistical analyses were performed using IBM SPSS Statistics for Windows, Version 27.0 (IBM Corp., Armonk, NY, USA). A *p*-value cut-off point of 0.05 at a 95% confidence interval (CI) was used to determine statistical significance. Survival analyses were carried out with the use of Kaplan-Meier curves and Cox regressions to calculate overall survival and disease-free survival. A univariate analysis (MANCOVA) was conducted using the Pearson chi-square test and independent-samples t-test to compare various descriptive variables with outcome measures. Post-hoc multinominal logistic regression analysis was used to assess the outcome measures according to the risk factors examined in the prediction model.

## 3. Results

“465 patients were found in the electronic registry that were treated for cSCC and qualified for the study, among which 72% were immunosuppressed (335), and the rest (130) were immunocompetent (controls). The baseline demographics between immunosuppressed patients and immunocompetent controls were similar in terms of ethnicity and religion; The Jewish population comprised 79% of the cohort (80% and 78% in the IS and control groups, respectively, *p* > 0.05), Muslims were the second biggest population (18% and 17% in the IS and control groups, respectively, *p* > 0.05), Christians and Druze comprised the remaining groups (no significant differences). Ashkenazi Jews (originated from America and Europe) were much of the cohort in both the IS and control groups (61% and 55%, respectively, *p* > 0.05), while the Sephardic Jews (originated from Asia and Africa) comprised the rest of both groups”.

Differences were seen in gender distribution. The percentage of men in the IS group was significantly higher (0.76) compared to immunocompetent controls (0.65, *p* = 0.02). As a group, IS patients did not differ significantly from immunocompetent controls in their TNM staging. Early disease (stages 1–2) compared to advanced disease (stages 3–4) were relatively similar in IS and control patients (69% and 74%, respectively, *p* > 0.05). In the immunosuppressed (IS) group, 124 patients were SOTR, 96 patients had hematopoietic malignancies, 42 had psoriasis, 40 had chronic kidney disease (CKD) and 33 had rheumatologic conditions (rheumatoid arthritis and lupus). Patients who were treated or had a diagnosis of more than one disease (apart from cSCC) were excluded from the study. Detailed information is depicted in Table 1.

Compared to the control group, IS patients demonstrated several differences. IS patients were younger at the time of cSCC diagnosis (68y), compared to controls (78y), *p* = 0.001. IS patients were also more likely to be men (75%), compared to controls (65%), *p* = 0.02. Higher rates of positive resection margins were noted in 20% of IS patients, compared to 9% in the control group (OR = 2.4, *p* = 0.007). The recurrence rate in the IS group was 24%, compared to 15% in controls (OR = 1.7, *p* = 0.04). Moreover, multiple cSCC tumors were seen in 28% of patients compared to 10% in the IS and control groups, respectively (OR = 3.6, *p* = 0.001). The average number of primary tumors in the IS group was 1.9, compared to 1.3 in the controls (*p* = 0.001). Post-operative external beam radiotherapy (with or without adjuvant chemotherapy) was given to 70% of IS patients compared to 27% of patients in the control group (*p* = 0.05), representing more aggressive tumors in the IS group. Patients in the IS group who were given immunomodulating drugs demonstrated worse survival rates (OR = 1.4, *p* = 0.04). The immunomodulating agents used in the cohort included Entracept, Rituximab, Fludarabine, and Cyclophosphamide. Cox regression analysis demonstrated statistically significant decreased overall survival (OS) rates for IS patients, compared to the controls (OR = 1.9, *p* = 0.031, 95% CI 1.6; 2.2). Detailed information is depicted in Table 1, Figure 1A.

Subgroup analysis within IS patients demonstrated significant differences depicted in Table 1, Figure 1B,C and Figure 2.

### 3.1. SOTR

Immunosuppressed SOTR patients begin their follow-up at a considerably younger age following transplant surgery (51y ± 4.2), compared to controls (78y ± 5.8) and to other IS patients (*p* < 0.05). Their cSCC diagnosis is also at a much younger age (60y ± 6.2, *p* < 0.001). These patients tend to have multiple malignant tumors (35%), with the highest number of primary tumors, compared to controls (2.54 tumors per patient on average, *p* < 0.001), but also compared to all other ID groups. The average SCC lesion size in the SOTR group was the smallest, measuring at 13.5 mm, compared to the control group and all other IS groups. High recurrence rates (27%) and a high percentage of positive margins on pathology specimens (23%), were also noted in SOTR patients. Within recurrent tumors, 6% were regional and 3% were distant. All-cause mortality for SOTR patients was significantly higher (23%), compared to controls (12%, *p* = 0.02) and other ID groups. Decreased survival rates were seen on Cox regression analysis compared to controls (HR = 2.4, *p* = 0.001), but also to all other ID groups. The 5-year disease-specific mortality hazard ratio was the highest compared to controls (HR = 2.8, *p* = 0.01).

### 3.2. CLL

Patients in the CLL group (n = 66) and LY group (n = 30) were mostly men (83% and 77%, respectively). CLL patients demonstrated the highest recurrence rates (28%), compared to controls (15%, *p* = 0.03) and all other IS patients. CLL patients also had the highest rates of positive margins (36%), compared to controls (9%, *p* < 0.01) and all other ID groups. They were also most likely to get adjuvant or definitive oncological treatments (36%), either radiotherapy or chemotherapy, compared to controls (15%, *p* = 0.02) and other ID groups. The 5-year disease-specific mortality hazard ratio was high compared to controls (HR = 2.6, *p* = 0.03).

### 3.3. CKD

Patients in the CKD group demonstrated the highest rates for multiple cSCC (OR = 4.7, *p* = 0.001) and the worst rates for overall survival on Cox regression analysis (HR = 3.2, *p* = 0.001). Detailed information about the different subgroups is depicted in Table 1.

### 3.4. RA and Psoriasis

Both rheumatoid arthritis and psoriasis patients demonstrated the shortest disease-free survival rates (2.9y ± 1.1 and 2.3y ± 0.7, respectively), compared to controls (4.1y ± 2.8) and to all other ID groups. Detailed information is available in Table 1, Figure 1B.

## 4. Discussion

The annual estimated incidence of cSCC in the US is 200,000–300,000 cases per year, or approximately 20% of all NMSCs [18]. Men after the age of 50 are at increased risk for cSCC, although there is evidence of an increase in incidence in younger age groups as well [19,20]. Most cases of cSCC are low-risk, with excellent overall survival rates following local surgical excision [21]. Nevertheless, Bachar et al. [22] found high-grade tumors and older age to be independent predictors of poor overall survival rates. Nodal metastasis in cSCC significantly impairs prognosis and occurs in 2–5% of patients [23,24,25]. Mizrachi et al. [26] demonstrated that the ratio between the number of positive metastatic nodes and the overall number of nodes dissected (i.e., the N-ratio) is a potentially valuable prognostic index in cutaneous squamous cell carcinoma.

Immunosuppression is strongly associated with increased risks of developing malignancies in general and specifically cSCC. Both the innate and adaptive immune system plays an important role in the development of cSCC, involving many elements of the immune system, including tumor-associated macrophages, dendritic cells, natural killer cells, and various cytokines [27]. Our understanding of these complicated mechanisms is far from complete. It is therefore no surprise that different immunosuppression conditions are associated with increased rates of cSCC [28]. From a clinical standpoint, the causes for immunosuppression can either be disease-related, in which different elements in the patient’s immune system are impaired, or treatment-related. In the latter, immunosuppression is acquired following pharmacological and oncological regimens.

In the present study, immunosuppressed patients were more likely to be men (0.76). Following the incidence rates and gender distribution of each IS disease, as reported in the literature, we expected the M:F ratio to be 0.6 (*p* = 0.01, calculations are not shown). This increase in cSCC in men relative to women was previously reported by Tam et al. [25] Other differences seen in the IS group were lower survival rates, increased odds for recurrence, multiple primary (malignant) tumors, and positive margins in pathological specimens, compared to the controls. They were also more likely to receive post-operative adjuvant therapy (radiotherapy +/− chemotherapy). Treatment with immunomodulating agents further decreases overall survival rates.

The most studied group of patients with acquired immunosuppression (treatment-related) are solid organ transplant recipients (SOTR). Different pharmacological agents are given for long periods of time following the transplant. The most known carcinogenic treatment frequently given to SOTR is cyclosporin [29]. Although SOTR are at increased risk for several malignancies, NMSC and specifically cSCC, account for approximately 40% of them [30,31]. Moreover, Ritter et al. [32] in a recent publication, reported worse outcomes in SOTR with cSCC compared to controls. In the present study, SOTR patients were diagnosed with cSCC at a significantly younger age and had the lowest survival rates among all other groups. Their average number of primary tumors was the highest, with a third of the patients having multiple (malignant) cutaneous tumors. Their average tumor size was significantly smaller compared to other groups. This is expected, as SOTR are prone to developing skin malignancies and consequently undergo repeated skin examinations. Skin malignancies are therefore likely to be diagnosed at early stages in these patients.

In our study, patients with SOTR were diagnosed with cSCC at a much younger age compared to all other groups. The aggressive nature of cSCC disease in these patients was seen in their higher tendency to develop multiple malignant tumors (2.54 tumors per patient on average), high recurrence rates, and high rates of positive margins on pathology specimens. All-cause mortality for SOTR patients was significantly higher compared to all other groups. The 5-year disease-specific mortality hazard ratio was the highest compared to controls (HR = 2.8, *p* = 0.01). Compared to all other groups, SOTR patients demonstrated worse outcomes in many (but not all) clinical and pathological variables that ultimately lead to an unfavorable course of disease and a poor prognosis.

Rheumatoid arthritis (RA) patients are another example of treatment-related immunosuppression, as they are frequently treated with either cyclosporin or tumor necrosis factor (TNF) antagonists. Wang et al. [33], published a systematic review and meta-analysis of 120,000 RA patients. Patients that were treated with anti-TNF drugs demonstrated an increased risk for non-melanoma skin cancers (NMSC). Relative risks for NMSC in general and specifically SCC were 1.28 (95% CI 1.19, 1.38) and 1.30 (95% CI 1.09, 1.54), respectively. The researchers concluded that RA patients treated with anti-TNF are at an increased risk of NMSC, especially SCC. In a similar report by Mercer et al. [34], assessed the risk for NMSC in a cohort of 15,000 RA patients that were treated with either anti-TNF or non-biological disease-modifying antirheumatic drugs (nbDMARD). Standardized incidence ratios (SIR) for NMSC were increased in both anti-TNF 1.72 (95% CI 1.43 to 2.04) and nbDMARD 1.83 (95% CI 1.30 to 2.50), compared with the general population.

The treatments for psoriasis often include ultraviolet A (UVA) light, which was previously reported as carcinogenic. Pouplard et al. [35], demonstrated increased risks for cutaneous squamous cell carcinoma (SIR = 5.3, 95% CI 2.63–10.71) among psoriasis patients that were treated with PUVA. Similar results were published by Boffetta et al. [14], and Ji et al. [36]. Increased risks for NMSC, including cSCC, were seen among hospitalized patients with psoriasis compared to the general population.

Aligned with these previous reports, both rheumatoid arthritis and psoriasis patients demonstrated the lowest disease-free survival rates (2.9y ± 1.1 and 2.3y ± 0.7, respectively), compared to controls (4.1y ± 2.8) and to all other IS groups. Our data further supports the conclusions made by the researchers mentioned in this section. It also highlights the importance of ongoing screening and regular follow-ups. More studies are needed to explore the relations between IS in RA and psoriasis patients and cSCC.

Other causes for immunosuppression are the inevitable course of some diseases (disease-related immunosuppression). One of the hallmarks of this is chronic lymphocytic leukemia (CLL). The incidence of CLL is constantly rising, with over 20,000 new patients diagnosed annually and a mortality rate of approximately 4000 a year in the United States (US) [37]. The risk for a second malignancy among CLL patients is increased, according to previous studies [38,39]. This phenomenon is explained, at least in part, by the immune deficiency seen in this disease [40]. CLL patients who receive immunomodulating chemotherapy are at even greater risk for second malignancies [41,42]. The most common second malignancies are non-melanoma skin cancers (NMSC); 37% of all second cancers, with a 2.3-fold increase in NMSC, compared to follicular lymphoma (FL) patients. Schollkopf et al. [43] reported 1105 s cancers among 12,373 CLL patients, with 39% of malignancies being NMSCs. A 3.66-fold increased risk of NMSCs was calculated when compared with the general population. Cutaneous SCC in CLL patients is much more aggressive than in immunocompetent patients, with increased risk for metastasis and recurrence, and unfavorable prognosis [44,45].

In the current study, CLL patients demonstrated a significantly higher TNM staging at presentation with an average tumor size of 27.4 mm. Their rates of recurrence, positive resection margins, and ECE rates were the highest compared to controls and all other groups. Our results support these previous publications on the aggressiveness of cSCC in CLL patients. It is recommended that patients with CLL undergo a full-body skin evaluation within 6 months of diagnosis [46].

Patients with chronic kidney failure (CKD) suffer from immunosuppression due to the accumulation of uremic toxins, which leads to chronic inflammation and oxidative DNA damage and increases the risk of cancer in general. One possible contributor to the immunosuppression in CKD patients, which ultimately leads to the development of malignancies, is the overall decreased function of naïve T cells and the accumulation of skewed regulatory versus T helper 17 ratios [47].

Only a small number of studies investigated the association between CKD and the development of cSCC. Wang et al. [48], conducted a nationwide, population-based study in Taiwan to assess the risk for non-melanoma skin cancers (NMSC) among patients with CKD. Their cohort included 1.5 million CKD patients undergoing regular hemodialysis (HD). In a nested case-control study, 80,000 CKD patients on HD with NMSC were matched to healthy controls. Among CKD patients there were 248 cases of NMSC occurred after a mean of 4.95 years of follow-up. The standardized incidence ratios (SIRs) for non-melanoma skin cancers (NMSC) CKD patients were 1.58 (95% confidence interval 1.39–1.79). The overall risk of NMSC in CKD (all stages), and stage 5 (HD patients) were 1.14-fold and 1.48-fold, respectively, compared with the general population. Of the patients on HD, a higher risk of NMSC was found in men (1.5-fold), South Taiwan residents (twofold), and patients with uremic pruritus after long-term antihistamine treatment (1.53-fold). The researchers concluded that CKD patients with chronic HD are at higher risk for developing NMSC and that uremic pruritis further increases the risk of NMSC.

Similar results were reported by Hurtlund et al. [49], in a nationwide study in Sweden and Denmark. The researchers assessed the risk for NMSC among 31,000 CKD patients undergoing regular HD. The SIR for overall cancer among CKD patients was 1.6 [95% CI, 1.5–1.6], and 5.3 (95% CI, 4.7–5.9) for NMSC specifically.

In our study, chronic kidney failure (CKD) was defined as a serum creatinine level above 1.5 mg/dL. Interestingly, 35% of patients with CKD had multiple cSCC, which was the highest among all other groups. Moreover, overall survival rates were the lowest for the patients in the CKD group compared to all other patients. With respect to the previous publications mentioned earlier, this study was the first to report the clinical characteristics as well as the natural course and prognosis of CKD patients with cSCC compared to other groups of immunosuppressed and immunocompetent patients. More studies are needed to explore the relationship between ID in CKD patients and cSCC.

Several potential pathogenetic mechanisms are believed to increase the risk for NMSC among autoimmune connective tissue diseases (ACTD) including SLE, such as disease-related impairment of the immune system, sustained cutaneous inflammation, drug-associated immune suppression, and increased susceptibility to acquired viral infections [16]. Gunawardane et al. [50], demonstrated an increased odds ratio of 1.5 (95% CI, 1.14–1.90) for developing cSCC among systemic lupus erythematosus (SLE) patients, compared to controls. Patients on immunosuppressive medication for at least one year had an OR of 1.69 (95% CI, 1.16–2.45) for developing non-melanoma skin cancer. With respect to Gunawardane et al., IS patients with SLE in our study did not demonstrate high-risk features of cSCC. The differences seen in this study may be explained by the relatively small sample size of SLE patients.

## 5. Limitations

Our cohort is retrospective, with its inherent limitations. One other limitation is the loss of follow-up among immunocompetent cSCC patients, who do not have a regular doctor treating them, as opposed to immunosuppressed patients with regular follow-ups in the different outpatient clinics in the hospital. Moreover, most of the immunocompetent controls choose to remove skin lesions in private outpatient clinics where the waitlist is much shorter. As a result, the overall “controls” in the hospital database are relatively small compared to the number of immunocompetent patients who continue regular follow-ups in hospital settings. For this reason, we believe that the number of immunocompetent cSCC patients treated in our medical center during the research years was higher than 130.

## 6. Conclusions

Our study focused on the effects of immunosuppression in cutaneous SCC patients. We focused on prognostic factors including survival, recurrences, and the natural course of the disease. Immunosuppressed patients demonstrate aggressive features including higher rates of PNI, ECE, and positive resection margins. The effects of the different conditions associated with immunosuppression in this study vary significantly. SOTR, CLL, CKD, RA, and psoriasis patients demonstrated worse outcomes compared to controls and should be therefore regarded as high-risk IS patients to develop cSCC.

## Figures and Tables

**Figure 1 cancers-15-01764-f001:**
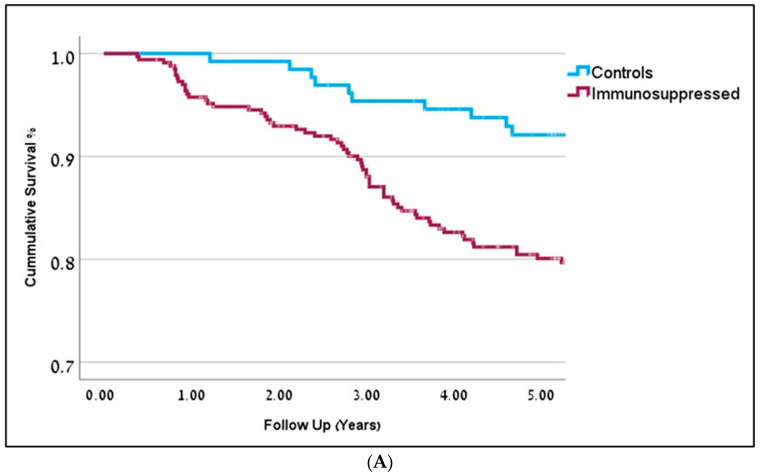
(**A**) Kaplan-Meier curves representing 5-year overall survival for immunosuppressed patients are in dark red (OR-0.81) and controls are in blue (OR = 0.91). Follow-up is represented in years (*p* = 0.001). (**B**) Kaplan-Meier curves representing 5-year overall survival (in years) for the different immunosuppression groups. Solid Organ Transplant Recipients (SOTR) in red; Chronic Lym-phocytic Leukemia (CLL) in bright green; Chronic Kidney Disease (CKD) in dark blue; Rheumatic Arthritis (RA) and Systemic Lupus Erythematosus (SLE) in purple; Psoriasis in green; Controls in blue. (**C**) Kaplan-Meier curves representing 5-year disease-free survival (in years) for the different immunosuppression groups. Solid Organ Transplant Recipients (SOTR) in red; Chronic Lym-phocytic Leukemia (CLL) in bright green; Chronic Kidney Disease (CKD) in dark blue; Rheumatic Arthritis (RA) and Systemic Lupus Erythematosus (SLE) in purple; Psoriasis in green; Controls in blue.

**Figure 2 cancers-15-01764-f002:**
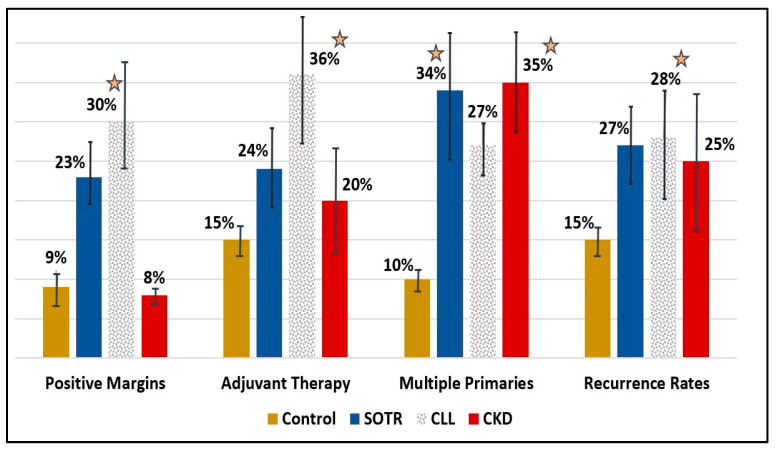
Bar-charts for different tumor characteristics (positive margins, adjuvant therapy, multiple primaries, and recurrence rates) for controls (black), SOTR (blue), CLL (polka dots), and CKD (red) patients. Asterisks represent a statistically significant difference (*p* < 0.05) compared to controls. Error bars represent the average result ± standard deviation. SOTR; Solid Organ Transplant Recipients, CLL; Chronic Lymphocytic Leukemia, CKD; Chronic kidney disease.

**Table 1 cancers-15-01764-t001:** Characteristics of the different immunosuppression groups and the immunocompetent controls.

	Controls	SOTR	CLL	LY	SLE	RA	Psoriasis	CKD
N (%)	130 (28%)	124 (27%)	66 (14%)	30 (6%)	13 (3%)	20 (4%)	42 (9%)	40 (9%)
Male	65%	79%	83%	77%	39%	50%	74%	78%
Age at exposure (Y)	78 ± 5.8	**51 ± 4.2**	73 ± 5.1	66 ± 3.6	69 ± 6.1	73 ± 2.7	69 ± 4.3	82 ± 5.4
Age at SCC diagnosis (Y)	78 ± 3.7	**60 ± 6.2**	75 ± 2.6	**68 ± 5.5**	69 ± 4.1	76 ± 6.3	71 ± 4.2	82 ± 43.3
Follow-up time (M)	53 ± 3.7	**36 ± 3.6**	50 ± 5.1	44 ± 6.7	44 ± 5.4	49 ± 4.4	46 ± 3.2	56 ± 7.2
Age at end of follow-up (Y)	83 ± 4.2	**65 ± 3.3**	79 ± 4.1	71 ± 2.7	73 ± 2.4	78 ± 3.2	75 ± 4.2	85 ± 4.6
All-cause mortality	0.12	**0.23**	0.2	0.22	0.18	0.19	0.12	**0.25**
Disease-free survival (M)	4.1 ± 2.8	3.1 ± 2.3	3.8 ± 2.7	4.2 ± 2.2	3.6 ± 1.2	**2.9 ± 1.1**	**2.3 ± 0.7**	2.8 ± 1.7
Size (mm)	23.9 ± 2.3	**13.5 ± 3.1**	27.4 ± 4.2	17.6 ± 3.3	**9.8 ± 2.8**	24 ± 4.4	22.8 ± 4.1	20.8 ± 3.6
Number of SCC	1.34	**2.54**	1.67	1.43	1.38	1.65	1.17	1.82
Recurrence	15%	27%	**28%**	20%	23%	25%	7%	25%
Multiple cSCC	10%	**35%**	27%	27%	15%	**30%**	10%	**35%**
Positive margins	9%	23%	**36%**	17%	15%	10%	2%	8%
PNI	5%	3%	9%	3%	0%	10%	0%	5%
ECE	2%	2%	9%	3%	0%	10%	0%	3%

Average numbers ± standard deviation (SD). Highlighted in bold and underlined are the statistically significant values (*p* < 0.05) compared to the controls. SOTR; Solid Organ Transplant Recipients, CLL; Chronic Lymphocytic Leukemia, LY; Lymphoma, RA; Rheumatic Arthritis, SLE; Systemic Lupus Erythematosus, CKD; Chronic Kidney Disease, PNI; Perineural Invasion, ECE; Extracapsular Extension, Y; years, M; months, mm; millimeters.

## Data Availability

The data that support the findings of this study are available from the corresponding author upon reasonable request.

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
