# Peer review of "Cutaneous Squamous Cell Carcinoma in Immunocompromised Patients—A Comparison between Different Immunomodulating Conditions"

_cancers, 2023, doi:10.3390/cancers15061764_

Round 1
Reviewer 1 Report
Dear Authors,
Thank you for submitting the article titled, “Cutaneous Squamous Cell Carcinoma in Immunocompromised Patients- a Comparison Between Different Immunomodulating Conditions“, written by Zavdy, et al. It looks that a certain level of significance can be seen, however, data are limited with respect to numbers of cases, region as well as results are similar to published article. Also, there are many unclear parts as follows:
1. The backgrounds of all patients are still unclear including races, stages of each disease, and history of medications. Therefore, there could be several items that caused selection bias.
2. What is the difference between Immunosuppression and Immunodeficiency (ID). Authors wrote “Immunosuppression” at the beginning and then used “Immunodeficiency (ID)” in the later part. Are these terms same or different?
3. What is the definition of Immunosuppression/Immunodeficiency (ID)? The criteria are unclear. Authors must explain them using appropriate markers such as leukocyte counts, CD4/CD8 T lymphocytes counts, and other immunity-related functions.
4. 465 patients look too small numbers for this size of hospital and for the period (about 10 years). Patients that are applicable to controls may be more than 131. Are there any patients that were excluded?
5. In the 3rd line from the top on page 4, authors describe “ID patients were also more likely to be men”. However, some diseases here usually may have more male patients than female patients. Are these results significant compared to the morbidity of each disease by sex?
6. In the 8th line from the top on page 4, authors describe “The average of number of primary tumors in the ID group…”. Are the tumor malignant, benign, or both?
7. In the 12th line from the top on page 4, authors describe “… who were given immunomodulating agents demonstrated …”. What kinds of immunomodulating agents were patients given? Please describe precisely.
8. Please put error bars to the graph in Figure 3.
9. Please add your institution’s ethical endorsement identification number to Materials and Methods.
Minor concerns:
1. There is a space between 4th and 5t line of the abstract. Please correct.
2. In the 3rd line of Introduction, authors describe “ofnonmelanoma skin cancer”. Please add a space.
3. Please add the summary of the abbreviations. What is “LY” in Table 1?
4. The time scale in Figure 1 is described as years, but that in Figure 2 and 3 is as months. It should be coordinated as same order.
Author Response
Reviewer 1
Thank you for submitting the article titled, “Cutaneous Squamous Cell Carcinoma in Immunocompromised Patients- a Comparison Between Different Immunomodulating Conditions“, written by Zavdy, et al. It looks that a certain level of significance can be seen, however, data are limited with respect to numbers of cases, region as well as results are similar to published article. Also, there are many unclear parts as follows:
- The backgrounds of all patients are still unclear including races, stages of each disease, and history of medications. Therefore, there could be several items that caused selection bias.
Reply: We appreciate the important remark made by the reviewer, and we added the following sentences in the results section:
"The baseline demographics between immunosuppressed patients and immunocompetent controls were similar in terms of ethnicity (the percentage of Jews versus non-Jews were similar in both groups). Differences were seen in gender distribution. The percentage of men in the IS group was significantly higher (0.76), compared to immunocompetent controls (0.65, p=0.02). As a group, IS patients did not differ significantly from immunocompetent controls in their TNM staging. Early disease (stages 1-2) compared to advanced disease (stages 3-4) were relatively similar in IS and control patients (69% and 74%, respectively, p>0.05)."
- What is the difference between Immunosuppression and Immunodeficiency (ID). Authors wrote “Immunosuppression” at the beginning and then used “Immunodeficiency (ID)” in the later part. Are these terms same or different?
Reply: We thank the reviewer for their remark. We used both terms interchangeably throughout the manuscript. To avoid further confusion, the term immunodeficiency was changed to immunosuppression across the article. Immunodeficient group ("ID group") was replaced by "IS group".
- What is the definition of Immunosuppression/Immunodeficiency (ID)? The criteria are unclear. Authors must explain them using appropriate markers such as leukocyte counts, CD4/CD8 T lymphocytes counts, and other immunity-related functions.
Replay: We thank the reviewer for their remark. The decision to refer to each of the conditions mentioned in the article as immunosuppression-associated (i.e. solid organ transplantees, chronic lymphocytic leukemia, chronic kidney disease, systemic lupus erythematosus, rheumatic arthritis and psoriasis) is based on extensive literature published elsewhere. All these conditions have been associated with immunosuppression according to many authors in the past. The pathophysiology of the intricate pathways to induce immunosuppression following each condition is different, and well beyond the scope of this research. We did not think it is in the nature of the current study to explain the data behind each condition, which would have taken the focus from the main goals of the current study. Nevertheless, we feel that further information on the different conditions was necessary and so changes have been made to the discussion.
Following the reviewer's comment, we amended the first paragraph of the discussion so to categorize IS as disease-associated (integral part of the pathophysiology of a specific condition) or treatment-associated (following immunomodulating therapies):
"Immunosuppression is strongly associated with increased risks for developing malignancies in general and specifically cSCC. Both innate and adaptive immune system play an important role in the development of cSCC, involving many elements of the immune system, including tumor-associated macrophages, dendritic cells, natural killer cells and various cytokines... From a clinical standpoint, the causes for immunosuppression can either be disease-related, in which different elements in the patient's immune system are impaired, or treatment-related. In the latter, immunosuppression is acquired following pharmacological and oncological regimens."
We continued the discussion by categorizing conditions that are associated with immunosuppression due to treatments (SOTR, RA, Psoriasis) and those that are an inevitable part in some conditions (CLL, CKD). We added specific parts in the discussion on the different theories and research on several potential pathogenetic mechanisms that are believed to increase the risk for both immunosuppression and malignancy in CKD and autoimmune connective tissue diseases (psoriasis being one of them).
- 465 patients look too small numbers for this size of hospital and for the period (about 10 years). Patients that are applicable to controls may be more than 131. Are there any patients that were excluded?
Reply: No patients were excluded from the research. We addressed the matter in the limitations section of the article:
"Our cohort is retrospective, with its' inherent limitations. One other limitation is the loss to follow-up among immunocompetent cSCC patients, who does not have a regular doctor treating them, as opposed to immunosuppressed patients with regular follow-ups in the different outpatient clinics in the hospital. Moreover, most of the immunocompetent controls choose to remove skin lesion in private outpatient clinics to which the waitlist in much shorter. As a results, the overall "controls" in the hospital database is relatively small, compared to immunocompetent patients who continue regular follow-ups in the hospital settings. For this reason, we believe that the number of immunocompetent cSCC patients treated in our medical center during the research years is higher than 130."
- In the 3rdline from the top on page 4, authors describe “ID patients were also more likely to be men”. However, some diseases here usually may have more male patients than female patients. Are these results significant compared to the morbidity of each disease by sex?
Reply: We appreciate the question. We calculated the expected gender distribution according to the incidence rates and gender distribution published in the literature (references are attached in the bottom of this page[1],[2],[3],[4],[5],[6],[7]). The expected M:F ratio was 0.6, when in fact the actual ratio in this study was 0.76 (p=0.01). This increase in men to women ration was reported in other studies. A clarification was added to the discussion:
In the present study, immunosuppressed patients were likely to be men (0.76). Following the incidence rates and gender distribution of each IS disease, as reported in the literature, we expected the M:F ratio to be 0.6 (p=0.01, calculations are not shown). This increase on cSCC men relatively to women was previously reported by Tam el al [25].
- In the 8thline from the top on page 4, authors describe “The average of number of primary tumors in the ID group…”. Are the tumor malignant, benign, or both?
Reply: Tumors in this article only refers to malignant ones. A clarification was added to the text.
- In the 12thline from the top on page 4, authors describe “… who were given immunomodulating agents demonstrated …”. What kinds of immunomodulating agents were patients given? Please describe precisely.
Reply: The immunomodulating agents used in the cohort included Entracept, Rituximab, Fludarabine and Cyclophosphamide. A clarification was added to the text.
- Please put error bars to the graph in Figure 3.
- Please add your institution’s ethical endorsement identification number to Materials and Methods.
Minor concerns (All were corrected):
- There is a space between 4thand 5t line of the abstract. Please correct.
- In the 3rdline of Introduction, authors describe “ofnonmelanoma skin cancer”. Please add a space.
- Please add the summary of the abbreviations. What is “LY” in Table 1? Lymphoma
- The time scale in Figure 1 is described as years, but that in Figure 2 and 3 is as months. It should be coordinated as same order.
[1] Ge F, Huang T, Yuan S, Zhou Y, Gong W. Gender issues in solid organ donation and transplantation. Ann Transplant. 2013;18:508–14.
[2] Catovsky, D., Wade, R., & Else, M. (2014). The clinical significance of patients' sex in chronic lymphocytic leukemia. Haematologica, 99(6), 1088–1094.
[3] Radkiewicz, C., Bruchfeld, J. B., Weibull, C. E., Jeppesen, M. L., Frederiksen, H., Lambe, M., Jakobsen, L., El-Galaly, T. C., Smedby, K. E., & Wästerlid, T. (2023). Sex differences in lymphoma incidence and mortality by subtype: A population-based study. American journal of hematology, 98(1), 23–30.
[4] Bikbov B, Perico N, Remuzzi G; on behalf of the GBD Genitourinary Diseases Expert Group : Disparities in chronic kidney disease prevalence among males and females in 195 countries: Analysis of the global burden of disease 2016 study. Nephron 139: 313–318, 2018
[5] Linos A, Worthington JW, O'Fallon WM, Kurland LT. The epidemiology of rheumatoid arthritis in Rochester, Minnesota: a study of incidence, prevalence and mortality. Am J Epidemiol. 1980;111:87–98.
[6] Fessel WJ. Systemic lupus erythematosus in the community. Incidence, prevalence, outcome and first symptoms; the high prevalence in black women. Arch Intern Med. 1974;134:1027–1035.
[7] Moll JM, Wright V. Psoriatic arthritis. Semin Arthritis Rheum. 1973;2:181–201.

Reviewer 2 Report
Reviewer's Comments to the Authors:
Article (cancers-2260473): Cutaneous Squamous Cell Carcinoma in Immunocompromised Patients- a Comparison Between Different Immunomodulating Conditions
Authors have done a comprehensive statistical analysis on the cutaneous squamous cell carcinoma (cSCC) patients, focusing on the following parameters: survival rates, higher recurrence rates, multiple primary tumors, high rates of positive margins and necessity for postoperative adjuvant therapy. The patients’ cohort consisted from the two groups: immunodeficient patients (composed of 7 different subgroups) and immunocompetent patients (served as control group). This is a nice piece of work, but requests some clarifications. I have suggested minor corrections, which, I hope, could be met and arranged by the authors. I hope this will improve the manuscript, thus it could be accepted for a publication.
The authors need to correct (or present a better explanation on) the following:
1. There is no need for the capital B in word Between in the title of the article.
2. It is necessary to include a few newer references throughout the paper, because only 2 out of 47 are from 2020 and 1 is from 2021!
3. Introduction part
- “There has been a constant increase in the incidence ofnonmelanoma skin cancer …” Correct the marked words.
- “Patients with CLL also exhibit an 8-fold increased risk for non-melanoma skin cancers [13]. Compared to the general population. Should be one sentence!
4. Results part
- Table 1 and results 3.2 section has LY abbreviation Average numbers ± standard deviation (SD) section bellow the Table 1 needs the explanation for LY abbreviation.
5. Discussion part
- “The incidence of CLL is constantly rising, with over 20,000 new patients diagnosed aannually and a mortality rate of approximately 4000 a year in the United States (US)” Correct the marked word. Also, there is no need for the full name of marked country since the abbreviation was used in the very first sentence in the discussion chapter “The annual incidence of cSCC in the US is estimated at…. “
- Any thoughts or reflections on: “With respect to Gunawardane et al., ID patients with SLE, did not demonstrate high-risk features of cSCC in this current study….. “ Please, comment did you observe any differences (in treatment of SLE, patients characteristics etc.) that might contribute to the different outcomes in these two studies!
- “One possible contributor to the immunodeficiency in CKD patients, that ultimately leads to the development of malignancies, is the overall decreased….” There is missing coma behind malignancies.
6. Acknowledgments: None.
- It would be nice if the authors thank all the patients who were treated in their hospital (what is voluntary act), since without data derived from their treatments they would not have the study!
Author Response
Reviewer 2
Authors have done a comprehensive statistical analysis on the cutaneous squamous cell carcinoma (cSCC) patients, focusing on the following parameters: survival rates, higher recurrence rates, multiple primary tumors, high rates of positive margins and necessity for postoperative adjuvant therapy. The patients’ cohort consisted from the two groups: immunodeficient patients (composed of 7 different subgroups) and immunocompetent patients (served as control group). This is a nice piece of work, but requests some clarifications. I have suggested minor corrections, which, I hope, could be met and arranged by the authors. I hope this will improve the manuscript, thus it could be accepted for a publication.
The authors need to correct (or present a better explanation on) the following:
- There is no need for the capital B in word Between in the title of the article. Corrected
- It is necessary to include a few newer references throughout the paper, because only 2 out of 47 are from 2020 and 1 is from 2021!
Reply: We appreciate the reviewer valuable remark. We amended the reference list accordingly and replaced some of the former articles that were listed. The current list has 15 out of 50 (30%) articles that were published between 2019-2023.
- Introduction part- “There has been a constant increase in the incidence ofnonmelanomaskin cancer …” Correct the marked words. Corrected.
- “Patients with CLL also exhibit an 8-fold increased risk for non-melanoma skin cancers [13]. Compared to the general population. Should be one sentence! Corrected
- Results part- Table 1 and results 3.2 section has LY abbreviationAverage numbers ± standard deviation (SD) section bellow the Table 1 needs the explanation for LY abbreviation. Corrected
- Discussion part- “The incidence of CLL is constantly rising, with over 20,000 new patients diagnosed aannuallyand a mortality rate of approximately 4000 a year in the United States (US)” Correct the marked word. Also, there is no need for the full name of marked country since the abbreviation was used in the very first sentence in the discussion chapter “The annual incidence of cSCC in the US is estimated at…. “ Corrected
- “With respect to Gunawardane et al., ID patients with SLE, did not demonstrate high-risk features of cSCC in this current study…..“ Please, comment did you observe any differences (in treatment of SLE, patients characteristics etc.) that might contribute to the different outcomes in these two studies!
Reply: We thank the reviewer's remark. We added the following sentence:
" The differences seen in this study may be explained by the relatively small sample of SLE patients."
- “One possible contributor to the immunodeficiency in CKD patients, that ultimately leads to the development of malignancies,is the overall decreased….” There is missing coma behind malignancies.
- Acknowledgments: None.
It would be nice if the authors thank all the patients who were treated in their hospital (what is voluntary act), since without data derived from their treatments they would not have the study!
Reply: We agree with this important remark. We added the following sentence:
"the authors wish to thank all the patients who were treated in our medical centre during the years 2012-2020 and made it possible for us to conduct this analysis and publish the results."

Reviewer 3 Report
I read with interest this manuscript regarding characteristics of cutaneous SCC in various patient groups based on various pathologies affecting relative immunity.
In its current form, I would recommend the following major changes to the manuscript to improve its scientific validity.
Major criticisms
1. The study population is unusual in that only 28% of patients are deemed immunocompetent controls. This is not reflective of usual practice where a much higher proportion of patients are not immunosuppressed. The authors do not comment on their population characteristics beyond stating the numbers, for eg, are they are specialist unit treating immunodeficient populations with “normal” patients being treated elsewhere? This is likely to introduce significant bias in comparisons.
2. Survival analysis here is with Overall Survival (OS) which is flawed given the co-morbidities in the immunodeficient group. A better measure would be the Disease Specific Survival (DSS) to account for SCC -related events only. If this cannot be done, then a discussion should be had of over interpreting the results.
3. The Kaplan Meier curves in Fig 1B, 1C and 1D are on different scales on both x- and y- axis making it difficult to read. Additionally, Fig 1 C and Fig 1D do not have the controls plotted making a comparison difficult.
4. CLL/LY : It is difficult to comment on the adjuvant or definitive oncological treatments here as it s unclear wether this is related to the SCC or to the CLL/LY. I am assuming the adjuvant Radiotherapy is SCC related, but the systemic treatment is unlikely to be base on the study period (2011-2020). The adjuvant and other treatments are combined as a single metric.
Minor criticisms
1. Introduction, paragraph 2, line 9 : “… non-melanoma skin cancers [13], compared to …”
2. Introduction, paragraph 3, line 1 : “This study aimed to assess the association of different…”
3. Table 1 : Acronym “LY” not introduced anywhere in manuscript - assumed to mean Lymphoma
4. Results, paragraph 1 : Immunodeficient population stated as 334 patients, with controls as 131. This differs from details in Table 1 (335 immunodeficient and 130 normal).
5. Fig 2 : Please provide error bars
Author Response
Reviewer 3
I read with interest this manuscript regarding characteristics of cutaneous SCC in various patient groups based on various pathologies affecting relative immunity. In its current form, I would recommend the following major changes to the manuscript to improve its scientific validity.
- The study population is unusual in that only 28% of patients are deemed immunocompetent controls. This is not reflective of usual practice where a much higher proportion of patients are not immunosuppressed. The authors do not comment on their population characteristics beyond stating the numbers, for eg, are they are specialist unit treating immunodeficient populations with “normal” patients being treated elsewhere? This is likely to introduce significant bias in comparisons.
Reply: We addressed the matter in the limitations section of the article:
"Our cohort is retrospective, with its' inherent limitations. One other limitation is the loss to follow-up among immunocompetent cSCC patients, who does not have a regular doctor treating them, as opposed to immunosuppressed patients with regular follow-ups in the different outpatient clinics in the hospital. Moreover, most of the immunocompetent controls choose to remove skin lesion in private outpatient clinics to which the waitlist in much shorter. As a results, the overall "controls" in the hospital database is relatively small, compared to immunocompetent patients who continue regular follow-ups in the hospital settings. For this reason, we believe that the number of immunocompetent cSCC patients treated in our medical center during the research years is higher than 130."
Survival analysis here is with Overall Survival (OS) which is flawed given the co-morbidities in the immunodeficient group. A better measure would be the Disease Specific Survival (DSS) to account for SCC -related events only. If this cannot be done, then a discussion should be had of over interpreting the results.
Reply: We reported on both the OS and DSS results in Fig 1B and 1C, accordingly.
The Kaplan Meier curves in Fig 1B, 1C and 1D are on different scales on both x- and y- axis making it difficult to read. Additionally, Fig 1 C and Fig 1D do not have the controls plotted making a comparison difficult.
Reply: The figures were corrected accordingly.
CLL/LY : It is difficult to comment on the adjuvant or definitive oncological treatments here as it s unclear wether this is related to the SCC or to the CLL/LY. I am assuming the adjuvant Radiotherapy is SCC related, but the systemic treatment is unlikely to be base on the study period (2011-2020). The adjuvant and other treatments are combined as a single metric.
Reply: Adjuvant radiotherapy relates only to the SCC treatment. A clarification was inserted in the text.
Minor criticisms (All were corrected)
1. Introduction, paragraph 2, line 9 : “… non-melanoma skin cancers [13], compared to …”
2. Introduction, paragraph 3, line 1 : “This study aimed to assess the association of different…”
3. Table 1 : Acronym “LY” not introduced anywhere in manuscript - assumed to mean Lymphoma
4. Results, paragraph 1 : Immunodeficient population stated as 334 patients, with controls as 131. This differs from details in Table 1 (335 immunodeficient and 130 normal).
5. Fig 2 : Please provide error bars (error bars were added)

Reviewer 4 Report
Overall well written clear and interesting observation.
Results section could be expanded upon. Not sure if the figures are in a standard format. BUt they are clear.
A small suggestion is to add bars to Fig 2 to indicate variance of the data. The figure doesn't seem to be in a standard format.
Author Response
Reviewer 4
Overall well written clear and interesting observation. Results section could be expanded upon. Not sure if the figures are in a standard format. BUt they are clear.
Reply: Both the results and the discussion were expanded.
A small suggestion is to add bars to Fig 2 to indicate variance of the data. The figure doesn't seem to be in a standard format.
Reply: Error bars were added.

Round 2
Reviewer 1 Report
Dear Authors,
Thank you for revising the manuscript “Cutaneous Squamous Cell Carcinoma in Immunocompromised Patients- a Comparison between Different Immunomodulating Conditions”, written by Zavday et al (cancers-2260473). The revision had been much improved, however, there is still a concern about patients’ backgrounds including the difference of races, as follow:
Major concern:
The patients’ backgrounds including the difference of races are still unclear. Authors describe “the percentage of Jews versus non-Jews were similar in both groups”. The patients’ racial background should include percentage of each race such as Caucasian, Negroid, Asian. Because the genetic background may have some influences on the results and also the readers resides all over the world.
Author Response
Reply: We thank the reviewer for their important observation. In terms of demographic characteristics, we reported on the age and gender differences between immunosuppressed patients and immunocompetent controls. We also reviewed the differences in ethnicity (Jews versus non-Jews), although we did not mention the percentages in the previous version, which we now revised again to include. In term of race, there is no distinction in the manuscript due to the specific nature of population in Israel. According to the Israeli Central Bureau of Statistics (CBS), the Israeli population in classified according to religion (Jews, Muslims, Christians and Druze), and according to descent (Ashkenazi Jews originate from America and Europe, while Sephardic Jews originate from Asia and Africa). There aren't further subclassifications of the population. These sentences were amended in the manuscript for further clarification:
"465 patients were found in the electronic registry that were treated for cSCC and qualified for the study, among which 72% were immunosuppressed (335), and the rest (130) were immunocompetent (controls). The baseline demographics between immunosuppressed patients and immunocompetent controls were similar in terms of ethnicity and religion; The Jewish population comprised 79% of the cohort (80% and 78% in the IS and control groups, respectively, p>0.05), Muslims were the second biggest population (18% and 17% in the IS and control groups, respectively, p>0.05), Christians and Druze comprised the remaining groups (no significant differences). Ashkenazi Jews (originated from America and Europe) were much of the cohort in both the IS and control groups (61% and 55%, respectively, p>0.05), while the Sephardic Jews (originated from Asia and Africa) comprised the rest of both groups".

Reviewer 3 Report
All queries addressed.
No further comments to review.
Author Response
We thank the reviewer for their important remarks.
Round 3
Reviewer 1 Report
Dear Authors,
Thank you for revising again the manuscript “Cutaneous Squamous Cell Carcinoma in Immunocompromised Patients- a Comparison between Different Immunomodulating Conditions”, written by Zavdy et al (cancers-2260473). The revision 2 had been much improved, and all concerns are now cleared.